# Are performance trajectories associated with relative age in French top 100 youth table tennis players? – A longitudinal approach

Irene R. Faber[1,2]*, Guillaume Martinent[3], Valérian Cece[3], Jörg Schorer[1]

**1** Institute of Sport Science, University of Oldenburg, Oldenburg, Germany, **2** International Table Tennis Federation, Lausanne, Switzerland, **3** Laboratory of Vulnerabilities and Innovation in Sport (EA 7428), University of Lyon, University of Claude Bernard Lyon 1, Lyon, France

* irene.faber@uol.de

**Data Availability Statement:** All data is open to the public through the website: http://www.fftt.com/site/competition/classement/classement-national.

## Abstract

Although relative age effects in sports have been studied worldwide, the underlying mechanisms are still under debate. This study adds to the existing knowledge by providing a further exploration of the association between relative age and the performance trajectories over four years in youth players of an individual skill/technique based sport: table tennis. Data of 1000 French male and female youth top 100 players across five ages (U14, U15, U16, U17 and U18) were collected from the ranking lists over a four-year period. A series of latent growth analysis was conducted per subsample and revealed three performance trajectories for male U14, U16 and U17 as well as for female U17 and U18 and four performance trajectories for male U15 and U18 and female U14, U15 and U16. Results of chi-square tests revealed that the players' birth quartiles were significantly associated with the performance trajectories only for male players U18 with a large effect size ($p = 0.01$; $W = .48$). All other male subsample only showed a trend for the male subsamples for those born in the fourth quartile. No relations or trends were found in the female subsamples. Future research in relative age effects should further explore individual characteristics and pathways while using a longitudinal approach in a prospective design and evaluate influencing constraints (and solutions) in a more comprehensive way.

## Introduction

The relative age effect (RAE) is described as a situation of inhomogeneous distribution of the players' birth dates within one age category. This means that the observed birth distribution differs from the expected one. Both within-year and between-year effects can be present as part of RAEs in a certain sport context [1–5]. Generally, within-year effects are described as deviated birth distributions per quartiles or semester and between-year effects as deviated birth distributions per year (i.e. per birth cohort within an age-category). In most sports, both effects are displayed as overrepresentations of the relatively older players who are born more early after the cut-off date compared to the relatively young players [2,3,5–8].

**Funding:** The author(s) received no specific funding for this work.

**Competing interests:** The authors have declared that no competing interests exist.

Studying RAEs in sports seems to have become an evergreen. Many authors have reported about the existence or absence of RAEs in many different kind of sports, while including the most likely/possible hypothesis that can explain their results [6,7,9,10]. The surplus of this kind of research is quite understandable for different reasons. First, RAEs in youth sports can lead to an unintended unfairness regarding sports participation, training, support and competition in sports [6,11]. This runs counter the principle of many to create equal chances for all children. Second, most national sports associations are urged by their government to keep up with the world's medal race [12]. For this purpose, they set-up talent programs to find and guide those young players that have the most potential to reach the (world's) elite level [13]. An unintended selection bias caused by relative age will cause an undesirable waste of potential (talented) youth players (false-negatives) and investments (false-positives) [6]. Third, RAEs appear to differ between sports, competition levels, sex, and age categories [14,15]. Hypothesizing about the existence of RAEs in especially youth sports and the underlying mechanism is not straightforward [16]. Conducting separate analyses per context and subsamples seems appropriate to unravel the etiology and influences in different sports since many factors play a role.

Wattie and colleagues (2015) proposed a theoretical framework to facilitate the understanding of RAEs in sports, which can be used as a starting-point when studying RAEs in a certain sport [16]. They based their framework on Newell's constraints-based model including three interacting types of constraints: individual, task and environmental constraints [17]. Suggested individual constraints within the framework are a player's birth date, physical maturation and size, sex and handedness/laterality [18–20]. Task constraints cover the type of sport and level of competitive play [6,7,15,21]. Cultural popularity, social norms, policies and development programs in sports and family influences are examples of the environmental constraints [2,22–25]. Besides these three types of constraints, the RAEs themselves are proposed as elements within the model that can interact bi-directionally with the constraints as soon as they exist within a context. The presence of more variations of RAEs (i.e. within-year and between-year effects) in a certain context, which even may interact, enhances the complexity of RAEs and their mechanisms even further [16]. Finally, change over time is also added as a component as all constraints can develop over time. Wattie et al. (2015) emphasize that relative age effects are the product of an interaction between an individual constraint and (task and) environmental constraints which develops over time [16]. A cascade of events in which the different elements interact, change the probability of the emergence and/or conservation of RAEs. The fact that most researchers only studied the RAEs by means of a cross-sectional design is a notable limit of this literature [6,7,9]. Hence, it seems highly unlikely to fully unravel the underlying mechanism of the relative age effects using only estimates based on one point in time. For that reason, it is important to evaluate RAEs and their influences while using a longitudinal approach.

To the best of our knowledge, no longitudinal studies have been published considering the association of relative age (effects) with youth players' performance trajectories competing in an individual and skill/technique based sport, in this case table tennis. Previously, mixed results have been presented in the (sub)samples of the cross-sectional studies in this sport [21,26–29]. These mixed results are hard to explain, but using a longitudinal approach could allow examining RAEs in more depth. In particular, examining the associations of quartiles' birth dates to the players' performance trajectories over time could provide further insight on RAEs. As such the aims of the present study were to: (a) examine first the cross-sectional within-year RAE effect as generally conducted in other studies (b) identify naturally-occurring performance trajectories of youth table tennis players and (c) examine whether youth players from distinct quartiles' birth dates belonged to particular identified performance trajectories.

Proposing hypotheses seems premature at this stage. In the existing cross-sectional literature, it is argued that relatively older players have better chances as they are generally more mature, further developed (e.g. physically and cognitively) and are more experienced [6,7]. However, on the long term the consequences might vary. Due to the better requisites, relatively older players are likely to be provided with the better opportunities for development (e.g. training and competition). This could lead to a reinforcement of the RAEs with an overrepresentation of the relatively older players especially on the higher levels. On the other hand, the relatively younger players that survive within the system might receive the better stimuli to develop their technical, tactical and mental skills and even turn-out to be the better performers on the long-run. [15,30] It is also suggested that once RAEs exist at youth levels and persist into developmental levels, relative age interacts with existing or new developmental constraints to produce new RAEs [16]. Thereof, we refrained from hypothesizing about the RAEs over time.

## Methods

### Design

A descriptive longitudinal approach was used to identify naturally occurring performance trajectories of French top 100 male and female youth table tennis players separately for each of the age categories (i.e. under 14 years (U14), U15, U16, U17 and U18) and to examine if players from distinct quartiles' birth dates belonged to particular performance trajectories. The study was conducted in compliance with the Declaration of Helsinki.

### Sample

Inclusion was based on the official national ranking lists of the FFTT of July 2017 (i.e. the end of the French table tennis competition season (2016–2017)). The French top 100 male and female players belonging to the U14, U15, U16, U17 and U18 were included in this study. The top 100 was identified as a relevant sample of the athletes involved by the talent identification in France. Considering the weaker opportunities and level of the athletes outside the top 100, it is expected that an expansion of the sample could biased an analysis focused on talent identification.

### Data collection & analysis

All data were recorded in anonymous data sets which were made available by the FFTT from their open archives. All data is open to the public through the website: http://www.fftt.com/site/competition/classement/classement-national. The birth month of each player was collected for analysis. These data were transferred into the accompanying birth quartile. The quartiles were determined periods of three months. Quartile 1 (Q1) represents the first period after the cut-off date (i.e. 1st of January) which covers the period from January to March. Quartile 2 (Q2) includes the second period from April to June, quartile 3 (Q3) the third period from July to September, and quartile 4 (Q4) the final period from October to December. In this study all players' data were analyzed while using the official cut-off dates for each age category. This also accounted for youth players competing in an older age category on the basis of excellent performance.

The players' performance levels were based on the competition rating scores provided by the FFTT [31]. It refers to a number of points allowing ranking all the French players. The higher the level of the opponent, the more points are earned following a win, whereas losses against players ranked below are penalized. Given that this performance score is actualized

half-yearly, eight measurement points (i.e., four-year period from August 2013 till July 2017) were used to ensure a comprehensive and sufficient period of time for conducting the statistical analyses. No athlete dropped in or dropped out among the measurements. The only missing data referred to athletes who were not classified in the French Federation of Table Tennis at the beginning of the data collection due to the presence of initial non-national players (n = 24, including 3 players in U14, 2 in U15, 2 in U16, 2 in U17, and 1 in U18 male samples; and 6 in U14, 2 in U15, 3 in U16, 2 in U17, and 1 in U18 female samples) or entry-to-practice after the first measurement point.

First, the observed birth date distributions per quartile and per birth cohort were calculated as percentage per the top 100 players (July 2017) of each subsample (i.e. per sex and age category). A chi-square analysis [3,32] including the calculation of the effect size (*W*) was used per subsample to test for main effects of the within-year effect (IBM SPSS Statistics 25; IBM Corp., Armonk, New York, United States of America). The actual birth distributions based on the French national birth statistics of the corresponding population (https://www.insee.fr/fr/statistiques/1893255) were used as references for the analysis.

Second, latent class growth analyses (LCGAs) were conducted to examine the longitudinal data using MPlus Version 7.3 (Los Angeles, CA, USA). LCGA is a statistical model which posits that an underlying grouping variable can be inferred from a set of indicators to discover distinct trajectories on a variable (performance scores in the present case) with different patterns of change and stability across time [33]. Ten sets of LCGAs were performed, one for each subsample (i.e., male and female U14, U15, U16, U17 and U18). In particular, a series of LCGA models was conducted to select models that precisely captured the shape and the number of the performance trajectories. Thus, a succession of models with increasing number of trajectories was achieved to identify which model was associated with the best-fit indices [34]. A mixture of statistical indices was used to identify the best-fitting model including the log-likelihood value, the Akaike information criterion (AIC), the Bayesian information criterion (BIC), the adjusted BIC (ABIC), and the Lo, Mendell, and Rubin likelihood ratio test (LRT). The smallest values of AIC, BIC, and ABIC and the highest log-likelihood scores designated the best-fitting model. Initial LCGA models included the mean level (intercept), linear and quadratic growths for each performance trajectory. Both LCGA models with linear and quadratic functions were compared with their respective LCGA models with only the linear function. The log likelihood ratio test allowed highlighting an eventual significant improvement of fit if fewer parameters were included in the model (i.e., omitting quadratic functions from LCGA models) [35].

Third, in order to examine whether players from particular performance trajectories belonged distinct quartiles' birth dates, we conducted a series of chi-square tests of association–performance profiles (3 or 4) × quartiles (4) including the calculation of the effect size (*W*) in IBM SPSS Statistics 25. Alpha was set on 0.05 for all analyses.

## Results

### Cross-sectional within-year effects

Table 1 presents the within-year birth distributions (quartiles) per subsample and the outcomes of the within-year analyses for female and male players. The observed birth distribution differed significantly from the French birth distribution for female U16 and for male U14, U16 and U18 with a medium effect size ($p < .05$; *W* ranged between .29 and .34). An underrepresentation in Q4 was present within the female U16 and male U14 subsamples, while male U16 showed an underrepresentation in Q3 and male U18 an overrepresentation of Q1. A trend of

**Table 1. Birth distribution per quartile for top 100 table tennis players according to age category.**

| Females | | | | | | | | |
|---|---|---|---|---|---|---|---|---|
| Age category | n | Q1 | Q2 | Q3 | Q4 | $\chi^2$ (3) | W | p |
| U14 | 100 | 25 | 26 | 28 | 21 | .88 | .09 | .83 |
| U15 | 100 | 33 | 20 | 28 | 19 | 6.08 | .25 | .11 |
| U16 | 100 | 22 | 33 | 32 | 13 | 9.87 | .31 | **.02** |
| U17 | 100 | 26 | 28 | 32 | 14 | 6.75 | .26 | .08 |
| U18 | 100 | 31 | 28 | 22 | 19 | 4.46 | .21 | .22 |
| Males | | | | | | | | |
| Age category | n | Q1 | Q2 | Q3 | Q4 | $\chi^2$ (3) | W | p |
| U14 | 100 | 35 | 30 | 20 | 15 | 11.43 | .34 | **.01** |
| U15 | 100 | 25 | 25 | 32 | 18 | 3.93 | .20 | .27 |
| U16 | 100 | 33 | 28 | 15 | 24 | 8.43 | .29 | **.04** |
| U17 | 100 | 27 | 24 | 30 | 19 | 2.47 | .16 | .48 |
| U18 | 100 | 37 | 21 | 20 | 22 | 9.43 | .31 | **.02** |

Q1: January-March, Q2: April-June, Q3: July-September, Q4: October-December.

underrepresentation of Q4 was recognized in the other subsamples (i.e. female U14, U17 and U18 and male U15 and U17), but only small non-significant effect sizes were found.

## Identification and characterization of performance trajectories

Results of LCGAs are presented in S1 Table. For female U14, U15 and U16, as well as male U15 and U18, results suggested that 3 and 4-class solutions could be retained. In particular, we observed big drops of AIC, BIC, and ABIC between 2 and 3 classes and between 3 and 4 classes. Additionally, LRT revealed that 4 classes fit better than 3 classes while 5 classes did not fit

**Table 2. Longitudinal performance trajectories across the 8 waves for the female subsamples.**

| Age category | Performance trajectories | Intercept | | | | Linear | | | Quadratic | | |
|---|---|---|---|---|---|---|---|---|---|---|---|
| | | N | Estimate | (SE) | p | Estimate | (SE) | p | Estimate | (SE) | p |
| U14 | High | 5 | 782.38 | (59.95) | < .01 | 176.02 | (17.41) | < .01 | -7.47 | (2.39) | < .01 |
| | Moderate-high | 21 | 646.37 | (23.26) | < .01 | 61.83 | (7.54) | < .01 | 2.77 | (0.93) | < .01 |
| | Moderate-low | 29 | 538.16 | (19.14) | < .01 | 34.83 | (8.07) | < .01 | 2.48 | (0.87) | < .01 |
| | Low | 45 | 513.04 | (5.52) | < .01 | 0.35 | (3.58) | 0.92 | 3.61 | (0.40) | < .01 |
| U15 | High | 4 | 1013.37 | (45.02) | < .01 | 189.02 | (17.84) | < .01 | -12.56 | (2.50) | < .01 |
| | Moderate-high | 21 | 730.08 | (20.44) | < .01 | 107.59 | (8.23) | < .01 | -3.07 | (1.20) | 0.01 |
| | Moderate-low | 38 | 644.23 | (12.18) | < .01 | 39.28 | (5.06) | < .01 | 1.77 | (0.84) | 0.03 |
| | Low | 37 | 543.01 | (8.59) | < .01 | 17.11 | (6.27) | < .01 | 2.83 | (0.79) | < .01 |
| U16 | High | 9 | 1242.45 | (39.79) | < .01 | 127.54 | (15.62) | < .01 | -8.51 | (2.24) | < .01 |
| | Moderate-high | 17 | 904.07 | (31.63) | < .01 | 114.44 | (9.80) | < .01 | -5.26 | (1.17) | < .01 |
| | Moderate-low | 31 | 736.82 | (23.36) | < .01 | 72.37 | (10.20) | < .01 | -1.54 | (1.13) | 0.17 |
| | Low | 43 | 582.97 | (12.27) | < .01 | 34.89 | (5.98) | < .01 | 2.09 | (0.88) | 0.02 |
| U17 | High | 11 | 1413.22 | (45.52) | < .01 | 100.87 | (12.00) | < .01 | -7.61 | (1.24) | < .01 |
| | Moderate | 31 | 943.73 | (27.99) | < .01 | 88.92 | (8.32) | < .01 | -4.48 | (0.88) | < .01 |
| | Low | 58 | 645.37 | (13.18) | < .01 | 49.08 | (5.89) | < .01 | -0.26 | (0.81) | 0.75 |
| U18 | High | 12 | 1596.56 | (31.14) | < .01 | 72.32 | (9.65) | < .01 | -6.34 | (1.37) | < .01 |
| | Moderate | 34 | 1050.92 | (24.16) | < .01 | 70.13 | (6.60) | < .01 | -5.51 | (0.79) | < .01 |
| | Low | 54 | 667.64 | (17.77) | < .01 | 47.55 | (5.54) | < .01 | -1.50 | (0.70) | 0.03 |

better than 4 classes for female U16. When comparing LCGA models, the substantive meaning of each of the emerging trajectories should be take into account in addition to consider the statistical indicators [33]. Thus, to achieve the balance between theoretical and statistical considerations, we used the model parameters to make sense of the classes and decide which model fits best [34]. Based on the interpretability of the performance trajectories and aforementioned statistical indicators, we selected a four-class solution for female U14, U15 and U16, as well as male U15 and U18. For female U17 and U18, as well as male U14, U16 and U17, we observed big drops of AIC, BIC, and ABIC between 2 and 3 classes. Additionally, LRT revealed that 3 classes fit better than 2 classes while 4 classes did not fit better than 3 classes for all the aforementioned subsamples. Hence, based on both the interpretability of the performance trajectories and the aforementioned statistical indicators, we selected a three-class solution for female U17 and U18, and male U14, U16 and U17. Finally, it is worth noting that the LRTs indicated significant worsening of fit if quadratic functions were omitted for all the subsamples (All LRTs > 12, $\Delta$df = 3 or 4, all $p_s \leq$ .01). Thus, both the linear and quadratic parameters were selected for all the ten examined subsamples.

The trajectories' estimates for the LCGA models and the number of participants belonging to each performance trajectory (i.e. high, moderate (high/low) and low) are presented in Table 2 and Fig 1 for female subsamples and in Table 3 and Fig 2 for male subsamples. As a whole, and as might be expected given the age of the participants, almost all the trajectories were characterized by a significant linear increase across time (17 out of the 18 trajectories for female and all the 17 trajectories for male). The so-called 'high' trajectories are characterized by the highest intercepts and the highest (or among the highest) linear slopes. In contrast, the 'low' trajectories present low levels of intercepts and linear slopes. Additionally, it is noteworthy that most of the trajectories were also characterized by significant quadratic parameters. In particular, 6 significant positive quadratic parameters (U shape over time) and 10 negative quadratic parameters (inverted U shape over time) emerged for female whereas 1 significant positive quadratic parameter and 14 negative quadratic parameters emerged for male.

The longitudinal performance trajectories across eight waves present an interesting variation of development. For the female athletes an interaction seems to happen for the U14 and U15 age categories, in which the higher performers are almost equal at the beginning, but then improve more across the eight waves. These differences are then retained over the years for the U16-U18 age groups. For the male athletes a slightly different pattern is found. While for the U14 the increase for the better performers is also higher, for the other age groups the differences remain the same or potentially even decrease with the years.

## Relationships between quartiles' birth dates and performance trajectories

The distribution of performance trajectories across the quartiles' birth dates as well as the results of chi-square tests are presented in Tables 4 and 5 for females and males, respectively. Results showed that quartiles' birth dates were significantly ($p$ = .01) related to performance trajectories among male U18 with a large effect size ($W$ = .48; Table 5). The 'high' performance trajectory shows a lower number of players in Q2 and Q4, while the 'low' trajectory seems to have an inverse RAE with an overrepresentation of the relatively younger ones. The 'moderate-high' trajectory reveals a typical RAE with an overrepresentation of the relatively older ones, and the 'moderate-low' trajectory presents no clear trend. However, it must be noted that the cells only include a small number. For all other analyses, no significant patterns were present with effect sizes ($W$) between .08 to .40. Only on a descriptive level it seems noteworthy that in all male age categories, except for males U16, no Q4 players were included in the 'high' performance trajectories, while no other empty cell were found (Table 5). Moreover, an

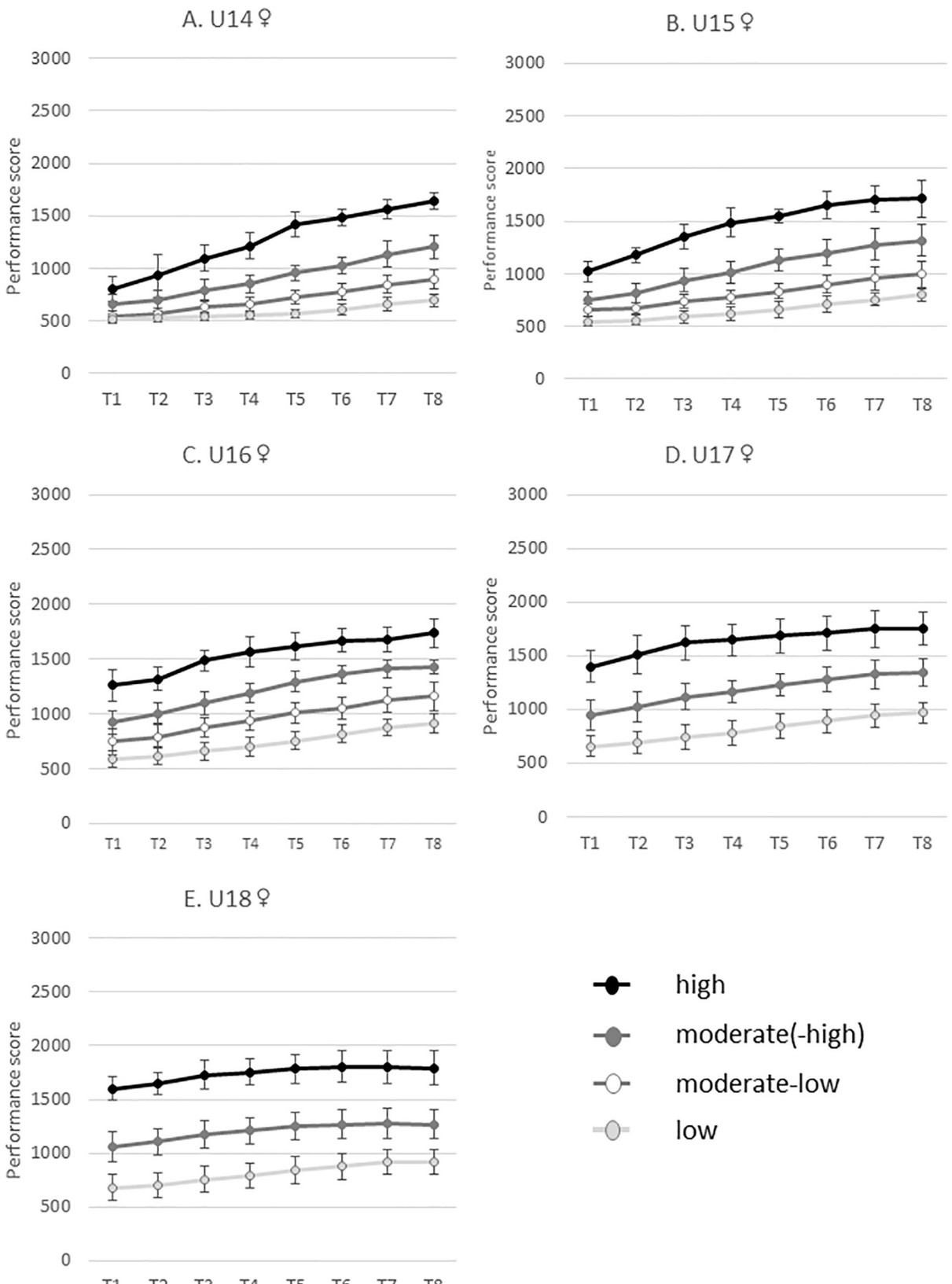

**Fig 1.** Longitudinal performance trajectories across the 8 waves (mean rating and standard deviation) for the female U14 (A), U15 (B), U16 (C), U17 (D) and U18 (E).

**Table 3. Longitudinal performance trajectories across the 8 waves for the male subsamples.**

| Age category | Performance trajectories | N | Intercept | | | Linear | | | Quadratic | | |
|---|---|---|---|---|---|---|---|---|---|---|---|
| | | | Estimate | (SE) | p | Estimate | (SE) | p | Estimate | (SE) | p |
| U14 | High | 6 | 1035.72 | (32.40) | < .01 | 251.06 | (12.11) | < .01 | -14.28 | (1.82) | < .01 |
| | Moderate | 32 | 761.96 | (183.27) | < .01 | 183.29 | (12.30) | < .01 | -5.67 | (1.42) | < .01 |
| | Low | 62 | 609.10 | (11.79) | < .01 | 77.90 | (6.18) | < .01 | 5.84 | (0.88) | < .01 |
| U15 | High | 5 | 1575.72 | (126.60) | < .01 | 188.68 | (41.52) | < .01 | -11.41 | (3.65) | < .01 |
| | Moderate-high | 19 | 1109.03 | (39.17) | < .01 | 181.15 | (17.20) | < .01 | -9.55 | (1.75) | < .01 |
| | Moderate-low | 42 | 797.91 | (31.71) | < .01 | 195.46 | (11.08) | < .01 | -9.83 | (1.34) | < .01 |
| | Low | 34 | 637.94 | (25.90) | < .01 | 118.02 | (11.84) | < .01 | 1.75 | (1.65) | 0.29 |
| U16 | High | 20 | 1573.59 | (45.34) | < .01 | 153.66 | (9.62) | < .01 | -10.36 | (1.02) | < .01 |
| | Moderate | 36 | 1236.81 | (30.70) | < .01 | 147.99 | (9.14) | < .01 | -7.68 | (0.95) | < .01 |
| | Low | 44 | 847.70 | (28.39) | < .01 | 176.78 | (10.02) | < .01 | -8.31 | (1.01) | < .01 |
| U17 | High | 19 | 1843.41 | (48.35) | < .01 | 127.55 | (11.48) | < .01 | -7.60 | (1.30) | < .01 |
| | Moderate | 43 | 1370.06 | (32.53) | < .01 | 142.25 | (9.84) | < .01 | -9.41 | (1.07) | < .01 |
| | Low | 38 | 993.78 | (37.07) | < .01 | 160.35 | (13.01) | < .01 | -7.23 | (1.56) | < .01 |
| U18 | High | 9 | 2008.86 | (65.93) | < .01 | 100.52 | (28.76) | < .01 | -6.26 | (1.36) | < .01 |
| | Moderate-high | 28 | 1587.05 | (36.44) | < .01 | 111.55 | (15.32) | < .01 | -7.36 | (1.20) | < .01 |
| | Moderate-low | 53 | 1307.08 | (23.48) | < .01 | 112.79 | (8.76) | < .01 | -6.99 | (0.90) | < .01 |
| | Low | 10 | 928.44 | (75.42) | < .01 | 94.79 | (29.40) | < .01 | 2.17 | (4.22) | 0.61 |

increasing pattern in the percentage of Q4 youth players from the 'high' to 'moderate' to 'low' trajectories appears in all male age categories. In the female subsamples no significant relations were find and no clear trends seemed present (Table 4).

## Discussion

This study used a longitudinal approach to investigate RAEs in more depth by examining whether youth table tennis players from distinct birth quartiles belonged to particular identified performance trajectories. The results of this study showed that only in the male youth players U18 a significant relationship ($p = .01$) with a large effect size ($W = .48$) was found between the birth quartiles and the performance trajectories; the higher performance trajectories (high and moderate-high) included relatively more older players compared to the lower performance trajectories (low-moderate and low). In addition to this, a trend was recognized in the other male subsamples of an underrepresentation of the relatively youngest ones (Q4) in the high performance trajectories compared to lower performance trajectories. In the females subsamples neither significant relationships nor clear trends were discovered.

The significant results within the male subsample U18 seems, at least to a certain extent, in line with the maturation-selection hypothesis [7], which complies with the interaction between the individual, task and environmental constraints [16]. The idea is that relative older youth players benefit from their physical advantages compared to their relatively younger peers [5]. These individual characteristics increase the chances of success and of being selected for a specific program. This mechanism can be reinforced by possible cognitive advantages in the relatively older youth players [5,36,37] and supportive psychological and sociocultural mechanisms [24].

The absence of relationships between the birth quartiles and the performance trajectories in the female subsamples and only the presence of 'Q4-trends' in the other males samples might be due to several factors. First, it looks like that most or all of the top 100 French youth players are offered the opportunities to develop themselves as performance increases within all

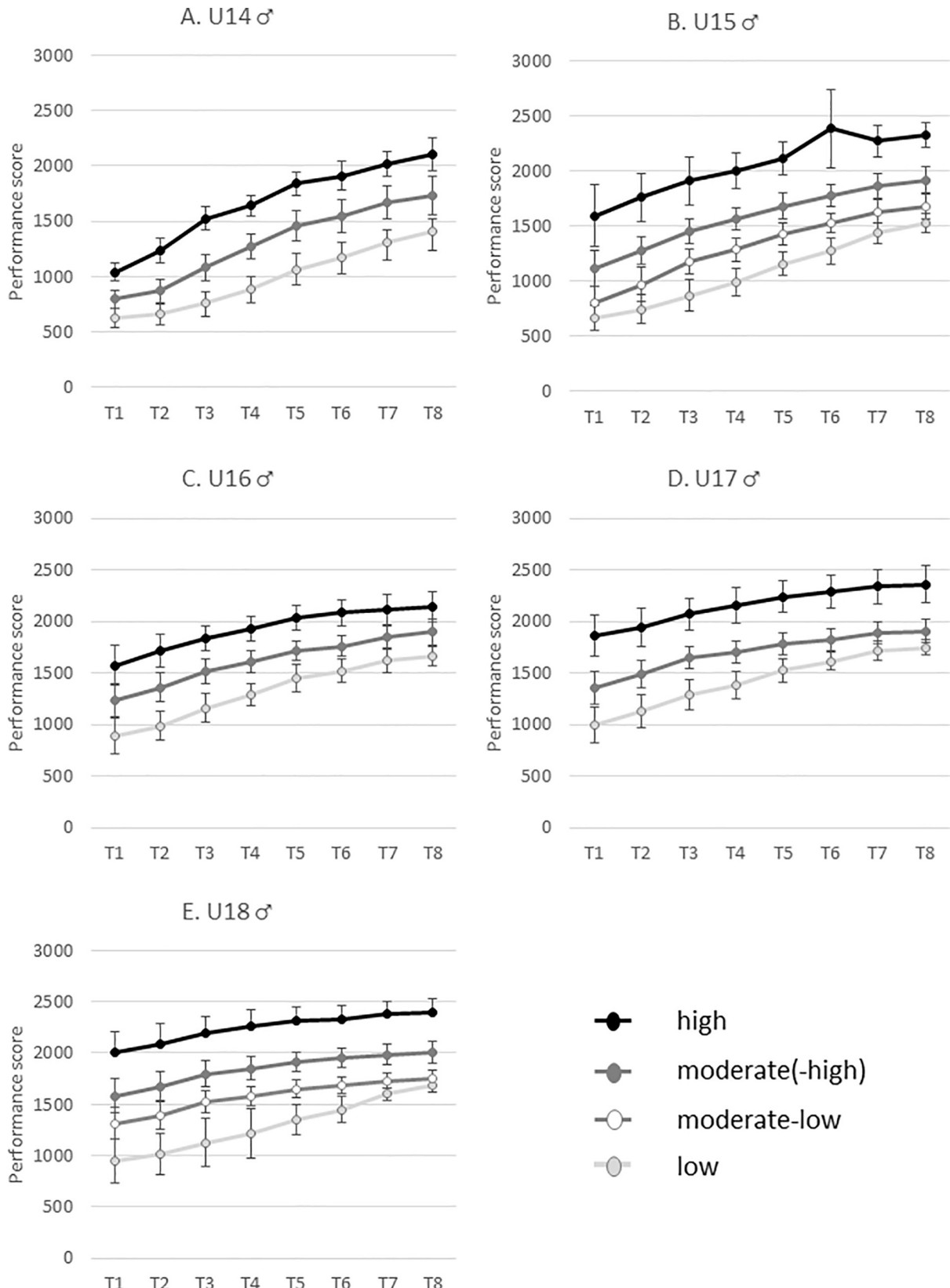

**Fig 2.** Longitudinal performance trajectories across the 8 waves (mean rating and standard deviation) for the male U14 (A), U15 (B), U16 (C), U17 (D) and U18 (E).

**Table 4. Relationships between quartiles' birth dates and performance trajectories of female players across the 5 subsamples.**

| Age category | Performance trajectories | Birth distribution per quartile | | | | | | | | $\chi^2$ | W | p |
|---|---|---|---|---|---|---|---|---|---|---|---|---|
| | | Q1 | | Q2 | | Q3 | | Q4 | | | | |
| U14 | High | 1 | 20.00% | 1 | 20.00% | 2 | 40.00% | 1 | 20.00% | 9.20 | 0.30 | .42 |
| | Moderate-high | 8 | 38.10% | 7 | 33.33% | 2 | 9.52% | 4 | 19.05% | | | |
| | Moderate-low | 5 | 17.24% | 10 | 34.48% | 8 | 27.59% | 6 | 20.69% | | | |
| | Low | 11 | 24.44% | 8 | 17.78% | 16 | 35.56% | 10 | 22.22% | | | |
| U15 | High | 1 | 25.00% | 1 | 25.00% | 1 | 25.00% | 1 | 25.00% | 2.40 | 0.15 | .98 |
| | Moderate-high | 7 | 33.33% | 3 | 14.29% | 7 | 33.33% | 4 | 19.05% | | | |
| | Moderate-low | 15 | 39.47% | 8 | 21.05% | 9 | 23.68% | 6 | 15.79% | | | |
| | Low | 10 | 27.03% | 8 | 21.62% | 11 | 29.73% | 8 | 21.62% | | | |
| U16 | High | 0 | 0.00% | 7 | 77.78% | 2 | 22.22% | 0 | 0.00% | 16.03 | 0.40 | .07 |
| | Moderate-high | 4 | 23.53% | 6 | 35.29% | 5 | 29.41% | 2 | 11.76% | | | |
| | Moderate-low | 7 | 22.58% | 12 | 38.71% | 9 | 29.03% | 3 | 9.68% | | | |
| | Low | 11 | 25.58% | 8 | 18.60% | 16 | 37.21% | 8 | 18.60% | | | |
| U17 | High | 6 | 54.55% | 2 | 18.18% | 3 | 27.27% | 0 | 0.00% | 10.93 | 0.33 | .09 |
| | Moderate | 9 | 29.03% | 10 | 32.26% | 10 | 32.26% | 2 | 6.45% | | | |
| | Low | 11 | 18.97% | 16 | 27.59% | 19 | 32.76% | 12 | 20.69% | | | |
| U18 | High | 4 | 33.33% | 4 | 33.33% | 2 | 16.67% | 2 | 16.67% | .61 | 0.08 | 1.00 |
| | Moderate | 11 | 32.35% | 9 | 26.47% | 7 | 20.59% | 7 | 20.59% | | | |
| | Low | 16 | 29.63% | 15 | 27.78% | 13 | 24.07% | 10 | 18.52% | | | |

Q1: January-March, Q2: April-June, Q3: July-September, Q4: October-December

**Table 5. Relationships between quartiles' birth dates and performance trajectories of male players across the 5 subsamples.**

| Age category | Performance trajectories | Birth distribution per quartile | | | | | | | | $\chi^2$ | W | p |
|---|---|---|---|---|---|---|---|---|---|---|---|---|
| | | Q1 | | Q2 | | Q3 | | Q4 | | | | |
| U14 | High | 2 | 33.33% | 2 | 33.33% | 2 | 33.33% | 0 | 0.00% | 6.82 | .26 | .34 |
| | Moderate | 13 | 40.63% | 12 | 37.50% | 3 | 9.38% | 4 | 12.50% | | | |
| | Low | 20 | 32.26% | 16 | 25.81% | 15 | 24.19% | 11 | 17.74% | | | |
| U15 | High | 1 | 20.00% | 3 | 60.00% | 1 | 20.00% | 0 | 0.00% | 11.46 | .34 | .25 |
| | Moderate-high | 7 | 36.84% | 7 | 36.84% | 4 | 21.05% | 1 | 5.26% | | | |
| | Moderate-low | 10 | 23.81% | 9 | 21.43% | 14 | 33.33% | 9 | 21.43% | | | |
| | Low | 7 | 20.59% | 6 | 17.65% | 13 | 38.24% | 8 | 23.53% | | | |
| U16 | High | 9 | 45.00% | 2 | 10.00% | 6 | 30.00% | 3 | 15.00% | 9.90 | .31 | .13 |
| | Moderate | 10 | 27.78% | 13 | 36.11% | 5 | 13.89% | 8 | 22.22% | | | |
| | Low | 14 | 31.82% | 13 | 29.55% | 4 | 9.09% | 13 | 29.55% | | | |
| U17 | High | 6 | 31.58% | 8 | 42.11% | 5 | 26.32% | 0 | 0.00% | 12.04 | .35 | .06 |
| | Moderate | 13 | 30.23% | 8 | 18.60% | 12 | 27.91% | 10 | 23.26% | | | |
| | Low | 8 | 21.05% | 8 | 21.05% | 13 | 34.21% | 9 | 23.68% | | | |
| U18 | High | 3 | 33.33% | 1 | 11.11% | 5 | 55.56% | 0 | 0.00% | 22.87 | .48 | **.01** |
| | Moderate-high | 16 | 57.14% | 5 | 17.86% | 5 | 17.86% | 2 | 7.14% | | | |
| | Moderate-low | 17 | 32.07% | 13 | 24.53% | 7 | 13.21% | 16 | 30.19% | | | |
| | Low | 1 | 10.00% | 2 | 20.00% | 3 | 30.00% | 4 | 40.00% | | | |

Q1: January-March, Q2: April-Jun, Q3: July-September, Q4: October-December

trajectories. Those youth players who managed to reach the top 100 can be part of the high, moderate or low trajectory and this seems, in general, not influenced by their relative age. Development seems thereof irrespective of a possible existing cross-section within-year effect. Second, it could be argued whether a focus on the national top levels (e.g. national top 10 or top 20) of consecutive birth cohorts, which are followed over the same development stage (U12 to U18) would provide different results. As the high performance trajectories in this study only included a small number of players (n = 4 to 20), maybe these are the ones that truly outperform their peer and represent the elite level with a strong competition [15]. Consequently, a separate analysis for this subsample is recommended with regard to the selection for talent development programs or national youth teams. Third, in addition to this, it could be due to the relatively small number of players that were actually included in the high performance trajectory, which challenges the power of the study. Fourth, it could be that the period involved per subsample and sex is of influence. As the subsamples cover a different time-span this might have caused differences in the results. It appears that in France (cross-sectional) RAEs start at the age of 14–15 years together with the selection for the 'big' events such as the youth European Championships. However, for U14, U15 or U16 in this study, the data gathering started when players were U12, U12 or U13, which might be too early to reveal an effect over time. Moreover, as confirmed earlier, RAEs differ between male and female subsamples, which might be related to the depth of competition in this case as the female French youth competition is considered as rather weak [15].

Another reasoning could be that individual characteristics and pathways need to be considered to fully understand the longitudinal mechanisms. For this study, it was probably implicitly assumed that all players within a particular quartile would develop similarly, because they are impacted by the same RAE. However, the players might have responded completely different to this effect, which is why no general effect is revealed. For this, Wattie and colleagues (2015) incorporated the principles of diversity and plasticity into their model [16,38,39]. These principles allow for inter-individual differences in development and reflect the potential for change across the lifespan, respectively. The fact that profiles of constraints vary between players, in combination with the interaction of all elements, results in unique trajectories of development. On this basis, Wattie et al. (2015) argued that each constraint represents a probabilistic causal mechanism for the emergence and/or maintenance of RAEs [16]. Thus, it can provide a unique constraint profile with context-specific configurations for a certain sport and/or individual, which may also include 'unknown constraints' that have not yet been revealed. Such profiles can be used to predict the likelihood of RAEs in certain contexts and/or test the impact of RAE interventions.

In addition to this, it seems worth to gain a deeper insight into the performance trajectories as well. When considering the performance curve of the female youth players it seems that the performance trajectories start quite closely together (Fig 2A(T1-T2)), diverge (Fig 2A(T3-T8), 2B) and consecutively have a stable distance between the performance curves (Fig 2C, 2D and 2E). It seems that the distances that exist around the age of 12 to 14 years remain over time in the top 100 girls. In contrast, for the male youth players a different patterns seems present. Although, similarly to the girls, they start closely together at the beginning of the performance trajectories and diverge over time (e.g. Fig 2A, 2B and 2C (T1-T4)), the curve converges again later on (e.g. Fig 2C (T6-T8), 2D (T5-T8), 2E (T3-T8)) and end rather closely together. An exception on the converging trend seems present in the high performance trajectory, which remains clearly at a higher position. The diverging-converging pattern in the French male top 100 youth players within the low and moderate curves might be due to the variation between adolescents in growth, maturation and development during this phase [40]. The exclusion from the high trajectory from the converging pattern is, too all probability, due to a

combination of the exclusiveness (i.e. high potential) of these players and the additional opportunities for (inter)national training and competition that are usually provided only to the nation's best players.

It is important to acknowledge two limitations of this study. First, the sample was included based on the official national French ranking lists of July 2017. From that point in time, performance data was retrieved from the previous years. Although it is expected that this 'retrospective' approach has not influenced the results to a large extent as composition of the top 100 seems quite stable, a future study with a prospective approach is recommended. Second, as is always the case with LCGA studies, the performance trajectories are data driven and sample specific. Hence, future research is needed to replicate the present findings with athletes from different countries or ages (e.g., professional athletes).

## Perspective

While the aim of this study was to look at the changes of relative age effects during the development of young elite table tennis players, there were no consistent patterns in birth quartile distributions in dependence of the trajectories. This might be partly because of the small sample of 100 players per age group, but perhaps more likely this might be due to the little exchange found between the varying trajectories. In this sample we look at elite players who are expected to remain in the talent development system given that France has thirteen intensive training centers.

Future studies should look at the changes in trajectories prior to being part of the national development systems to see, if these changes of birth quartile distribution could happen most probably before any selection process happens. Additionally, this might also be a first hint, that only if you are within this developmental system, you can actually reach this level of competition and there is probably no alternative route to success in French table tennis. Therefore, future studies should look at sports, in which more than one route exists. This might help us how relative age effects actually develop over time [16].

## Supporting information

**S1 Table. Fit indices of Latent Class Growth Analysis (LCGA) models with 1–5 classes for subsamples.**
(PDF)

## Acknowledgments

We acknowledge the French Table Tennis Federation (FFTT) for their support in data-collection.

## Author Contributions

**Conceptualization:** Irene R. Faber, Guillaume Martinent, Valérian Cece, Jörg Schorer.

**Data curation:** Guillaume Martinent.

**Formal analysis:** Irene R. Faber, Guillaume Martinent, Valérian Cece.

**Methodology:** Irene R. Faber, Guillaume Martinent, Valérian Cece.

**Project administration:** Irene R. Faber.

**Supervision:** Irene R. Faber.

**Visualization:** Irene R. Faber, Guillaume Martinent, Valérian Cece.

**Writing – original draft:** Irene R. Faber.

**Writing – review & editing:** Irene R. Faber, Guillaume Martinent, Valérian Cece, Jörg Schorer.

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
