## [Decision Letter · Decision Letter 0]

9 Jan 2020

PONE-D-19-33691

Are performance trajectories associated with relative age in French top 100 youth table tennis players? – A longitudinal approach

PLOS ONE

Dear Dr. Faber,

Thank you for submitting your manuscript to PLOS ONE. After careful consideration, we feel that it has merit but does not fully meet PLOS ONE’s publication criteria as it currently stands. Therefore, we invite you to submit a revised version of the manuscript that addresses the points raised during the review process.

Please, address the issues raised by Reviewer 1 and revise your paper based on Reviewer 2 comments.

We would appreciate receiving your revised manuscript by Feb 23 2020 11:59PM. To enhance the reproducibility of your results, we recommend that if applicable you deposit your laboratory protocols in protocols.io, where a protocol can be assigned its own identifier (DOI) such that it can be cited independently in the future. For instructions see: http://journals.plos.org/plosone/s/submission-guidelines#loc-laboratory-protocols

We look forward to receiving your revised manuscript.

Kind regards,

Gábor Vattay, PhD, DSc

Academic Editor

PLOS ONE

Journal Requirements:

Reviewers' comments:

Reviewer's Responses to Questions

**Comments to the Author**

1. Is the manuscript technically sound, and do the data support the conclusions?

Reviewer #1: Yes

Reviewer #2: Yes

2. Has the statistical analysis been performed appropriately and rigorously? 

Reviewer #1: Yes

Reviewer #2: Yes

3. Have the authors made all data underlying the findings in their manuscript fully available?

Reviewer #1: Yes

Reviewer #2: Yes

4. Is the manuscript presented in an intelligible fashion and written in standard English?

Reviewer #1: Yes

Reviewer #2: Yes

5. Review Comments to the Author

Reviewer #1: The study investigates the association between relative age and the performance trajectories over four years in youth table tennis players. The authors found that the players’ birth quartiles were significantly associated with the performance trajectories only for male players U18 with a large effect size (p = 0.01; W = .48). All other male subsample only showed a trend for the male subsamples for those born in the fourth quartile. No relations or trends were found in the female subsamples.

The results of this study are in my opinion of relevance to table tennis. However this study, as it stands, does not concisely focus on those aspects that would argue in favor of the necessity and novelty of this research. It could have an interest to a small, specific readership. In my opinion, the paper is more suited to the journals specializing in sport and exercise sciences.

Reviewer #2: Plos One

Review: Are performance trajectories associated with relative age in French top 100 youth table tennis players? – A longitudinal approach

General Comments

This manuscript examines performance trajectories associated with relative age in French top 100 youth table tennis players.

This study adds knowledge to the existing literature by providing a further exploration of the association between relative age and the performance trajectories over four years in youth players of an individual skill/technique. From my point of view, this a great research idea that should be extended to other sports, in particular to sports based on strength.

However, in the current version, I have some major concerns. In the following, I have listed my suggestions.

Major suggestions:

1) Please clarify in the whole manuscript, in particular in the abstract that you did not perform a classic longitudinal approach. I guess the athletes in the top 100 do not stay over 4 years. In other words please describe how many athletes stayed in the sample over 4 years and how many dropped in/dropped out

2) A large part of the paper is about Latent Class Growth Analysis (LCGA) models. I would recommend just to provide the final results and stick to your main results

3) Please comment why you did take only the top 100 athletes. It would be a major improvement if you would/could increase your sample.

6. PLOS authors have the option to publish the peer review history of their article (what does this mean?). If published, this will include your full peer review and any attached files.

Reviewer #1: No

Reviewer #2: No

---

## [Author Response · Author response to Decision Letter 0]

23 Jan 2020

Point-by-point response

Response:

PLOS ONE’s style requirement have been checked carefully. Changes are made in the lay-out regarding the title page, headings, tables and page numbers. We did not present these changes using the track-changes option for readability. We hope to meet all requirements now. 

Reviewer #1: The study investigates the association between relative age and the performance trajectories over four years in youth table tennis players. The authors found that the players’ birth quartiles were significantly associated with the performance trajectories only for male players U18 with a large effect size (p = 0.01; W = .48). All other male subsample only showed a trend for the male subsamples for those born in the fourth quartile. No relations or trends were found in the female subsamples.

The results of this study are in my opinion of relevance to table tennis. However this study, as it stands, does not concisely focus on those aspects that would argue in favor of the necessity and novelty of this research. It could have an interest to a small, specific readership. In my opinion, the paper is more suited to the journals specializing in sport and exercise sciences. 

Response

Thank you for your careful considerations. As stated in the introduction, hypothesizing about the existence of RAEs in several contexts but especially youth sports and the underlying mechanism is not straightforward. Conducting separate analyses per context and subsamples seems appropriate to unravel the aetiology and influences in different sports since many factors play a role. Moreover, there is only limited literature available that focuses on the development over time. Most researchers only studied the RAEs by means of a cross-sectional design which is a notable limit of this literature. It seems highly unlikely to fully unravel the underlying mechanism of the relative age effects using only estimates based on one point in time. For that reason, to our opinion it is important to evaluate RAEs and their influences while using a longitudinal approach. To the best of our knowledge, no longitudinal studies have been published considering the association of relative age (effects) with youth players’ performance trajectories competing in an individual and skill/technique based sport. This study intends to add knowledge to the existing literature by providing a further exploration of the association between relative age and the performance trajectories over four years in youth players of an individual skill/technique with the ultimate aim to create fairness regarding sports participation, training, support and competition in youth sports. 

Reviewer #2: Plos One

Review: Are performance trajectories associated with relative age in French top 100 youth table tennis players? – A longitudinal approach

General Comments

This manuscript examines performance trajectories associated with relative age in French top 100 youth table tennis players.

This study adds knowledge to the existing literature by providing a further exploration of the association between relative age and the performance trajectories over four years in youth players of an individual skill/technique. From my point of view, this a great research idea that should be extended to other sports, in particular to sports based on strength.

However, in the current version, I have some major concerns. In the following, I have listed my suggestions.

Major suggestions:

1) Please clarify in the whole manuscript, in particular in the abstract that you did not perform a classic longitudinal approach. I guess the athletes in the top 100 do not stay over 4 years. In other words please describe how many athletes stayed in the sample over 4 years and how many dropped in/dropped out

Response

For each sub-sample, the population refers to the top 100 in the last measurement point (T8 - July 2017). This information is specified within the sample section: “Inclusion was based on the official national ranking lists of the FFTT of July 2017 (i.e. the end of the French table tennis competition season (2016-2017)). The French top 100 male and female players belonging to the U14, U15, U16, U17 and U18 were included in this study”. For the longitudinal analyses, these athletes were monitored during the data collection (the four previous years). During this period, most of the athletes have maintained their top 100 position among the seasons of the collections. Consequently, no athlete dropped in or dropped out among the measurements. The only missing data referred to athletes who were not classified in the French Federation of Table Tennis at the beginning of the data collection due to the presence of initial non-national players (n = 24, including 3 players in U14, 2 in U15, 2 in U16, 2 in U17, and 1 in U18 male samples; and 6 in U14, 2 in U15, 3 in U16, 2 in U17, and 1 in U18 female samples) or entry-to-practice after the first measurement point. This information is added to the method section >> Data collection & analysis (line 156-161). 

2) A large part of the paper is about Latent Class Growth Analysis (LCGA) models. I would recommend just to provide the final results and stick to your main results

Response

We acknowledge that the paper focused on a large part on latent class growth analysis methodology and it could be confusing for the reader. The previous table 2 (fit indices) will thus be attached in supplemental materials (S1 Table). However, to our point of view, the trajectories description (previous tables 3 and 4; new table 1 and 3) and the relationships between quartiles’ birth dates and trajectories (previous tables 5 and 6; new table 4 and 5) are necessary to address the aims of the present study.

3) Please comment why you did take only the top 100 athletes. It would be a major improvement if you would/could increase your sample. 

Response

The top 100 was identified as a relevant sample of the athletes involved by the talent identification in France. Considering the weaker opportunities and level of the athletes outside the top 100, we consider that an expansion of the sample could biased an analyse focused on talent identification. This information is added to the method section >> Sample (line 132-135).

---

## [Decision Letter · Decision Letter 1]

6 Apr 2020

Are performance trajectories associated with relative age in French top 100 youth table tennis players? – A longitudinal approach

PONE-D-19-33691R1

Dear Dr. Faber,

We are pleased to inform you that your manuscript has been judged scientifically suitable for publication and will be formally accepted for publication once it complies with all outstanding technical requirements.

With kind regards,

Gábor Vattay, PhD, DSc

Academic Editor

PLOS ONE

Additional Editor Comments (optional):

Reviewers' comments:

Reviewer's Responses to Questions

**Comments to the Author**

1. If the authors have adequately addressed your comments raised in a previous round of review and you feel that this manuscript is now acceptable for publication, you may indicate that here to bypass the “Comments to the Author” section, enter your conflict of interest statement in the “Confidential to Editor” section, and submit your "Accept" recommendation.

Reviewer #1: (No Response)

2. Is the manuscript technically sound, and do the data support the conclusions?

Reviewer #1: Yes

3. Has the statistical analysis been performed appropriately and rigorously? 

Reviewer #1: Yes

4. Have the authors made all data underlying the findings in their manuscript fully available?

Reviewer #1: Yes

5. Is the manuscript presented in an intelligible fashion and written in standard English?

Reviewer #1: Yes

6. Review Comments to the Author

Reviewer #1: The revised version of the manuscript has improved. However, findings of this study are of relevance for a specific group of table tennis players. As such it could have an interest to a small, specific readership. In my opinion, the paper is more suited to the journals specializing in sport and exercise sciences.

7. PLOS authors have the option to publish the peer review history of their article (what does this mean?). If published, this will include your full peer review and any attached files.

Reviewer #1: No

---

## [Editor Report · Acceptance letter]

9 Apr 2020

PONE-D-19-33691R1 

Are performance trajectories associated with relative age in French top 100 youth table tennis players? – A longitudinal approach 

Dear Dr. Faber:

I am pleased to inform you that your manuscript has been deemed suitable for publication in PLOS ONE. Congratulations! Your manuscript is now with our production department. 

With kind regards,

on behalf of

Dr. Gábor Vattay 

Academic Editor

PLOS ONE